# An Origami Paper-Based Device Printed with DNAzyme-Containing DNA Superstructures for *Escherichia coli* Detection

**DOI:** 10.3390/mi10080531

**Published:** 2019-08-12

**Authors:** Yating Sun, Yangyang Chang, Qiang Zhang, Meng Liu

**Affiliations:** 1School of Environmental Science and Technology, Key Laboratory of Industrial Ecology and Environmental Engineering (Ministry of Education), Dalian University of Technology, Dalian 116024, China; 2School of Bioengineering, Dalian University of Technology, Dalian 116024, China

**Keywords:** colorimetric, DNAzyme, *Escherichia coli*, paper-based device, rolling circle amplification

## Abstract

Rapid detection of pathogenic bacteria is extremely important for public health and safety. Here, we describe for the first time an integrated origami paper-based analytical device (PAD) incorporating cell lysis, molecular recognition, amplification and visual detection of *Escherichia coli* (*E. coli*). The device features three components: paper for its ability to extract protein molecules nonspecifically from cells, DNA superstructures for their ability to immobilize RNA-cleaving DNAzymes (RCDs) but undergo target-induced RNA cleavage on paper, and isothermal rolling circle amplification (RCA) for its ability to amplify each cleavage event into repetitive sequence units that can be detected by naked eye. This device can achieve detection of *E. coli* K12 with a detection limit of as low as 10^3^ CFU·mL^−1^ in a total turnaround time of 35 min. Furthermore, this device allowed the sensitive detection of *E. coli* in complex sample matrices such as juice and milk. Given that more specific RCDs can be evolved for diverse bacteria, the integrated PAD holds great potential for rapid, sensitive and highly selective detection of pathogenic bacteria in resource-limited settings.

## 1. Introduction

In recent years, infectious diseases caused by food- and water-borne bacterial pathogens have a high incidence throughout the world [1,2,3,4]. The World Health Organisation (WHO) estimates that 600 million foodborne illnesses and 420,000 deaths were reported in 2010. In addition, about 30% of all foodborne deaths are children under 5 years [5]. According to the World Health Organization, diarrhea pathogens, such as *Escherichia coli, Salmonella, Helicobacter pylori*, *Campylobacter* and *Vibrio cholerae* [5,6,7,8], account for about one-third infections. Among these pathogenic bacteria, the main cause of diarrhea is *Escherichia coli* (*E. coli*) [9]. *E. coli* is a widely distributed Gram-negative bacterium that mainly lives in the intestines of humans and warm-blooded animals for facultative anaerobic activities. However, it can also cause serious symptoms such as sepsis and hemolytic uremic syndrome [10,11], and even result in serious diseases including diarrhea, gastroenteritis, inflammation and malnutrition. The U.S. EPA regulation requires that no *E. coli* pathogens are allowed in drinking water. Since *E. coli* can cause pathogenicity at low doses (most likely less than 100 organisms) [12], the development of a rapid, sensitive, low-cost and highly specific method for identifying *E. coli* is of great significance in the fields of medical diagnostics, food safety, and environmental monitoring.

Traditional culture-based methods for qualitative and quantitative detection of pathogens are powerful. However, the requirement for pre-enrichment in non-selective culture broth, selective enrichment and plating, biochemical analysis and serological confirmation makes these methods labour-intensive and time-consuming, as the entire process can take several days [13,14]. In recent years, many efforts have been directed toward the development of new methods or techniques including immuno-based assays [15,16], polymerase chain reaction (PCR) [17], surface plasmon resonance (SPR) [18], flow cytometry (FC) [19] and loop-mediated isothermal amplification (LAMP) [20,21,22]. However, these methods entail multiple processing steps, hence increasing the risk of cross-contamination. Furthermore, expensive equipment and highly trained personnel are required, thus limiting the use of this method in resource-limited settings.

Recent advances in functional nucleic acids (FNAs) have enabled the use of them as molecular recognition elements for various bioanalytical and biomedical applications [23,24]. FNAs, mainly include DNA and RNA aptamers, deoxyribozymes (DNAzymes) and ribozymes (RNA based enzymes), are small fragments of nucleic acid molecules with defined structures and functions (such as binding and/or catalysis). Aptamers and DNAzymes have not been found in natural systems but they can be isolated from random-sequence libraries of nucleic acids via “systematic evolution of ligands by exponential enrichment (SELEX)”. Among these catalytically active DNA molecules, one well-known DNAzyme is the RNA-cleaving DNAzyme (RCD) that can efficiently catalyze the RNA cleavage reaction [25]. Recently, we have shown that RCDs can be activated by a specific pathogen, such as *E. coli* and *Clostridium difficile* [12,26]. Furthermore, these RCDs have been coupled to diverse DNA amplification strategies to allow the development of optical biosensors for the detection of bacteria. In these biosensing systems, the sample pre-treatment including cell lysis and protein extraction, DNA amplification and detection have been separately performed, thus restricting their application for point-of-care (POC) testing. Recently paper has been widely used for engineering affordable, user-friendly, portable, rapid and robust devices, thus holding great potential to deliver POC testing to resource-limited settings. Several paper-based analytical devices (PADs) have been demonstrated for the detection of a wide range of analytes [27,28,29]. However, functionalization of paper with FNAs such as DNAzymes has not been exploited for the design of fully integrated PADs for monitoring of pathogenic bacteria. 

In this study, we developed a disposable origami paper-based analytical device (denoted as oPAD) that fully integrates on-paper cell lysis for protein extraction, target-responsive RCDs for molecular recognition, rolling circle amplification (RCA) for signal amplification, origami technique for reagent diffusion and flow, and portable readout. For this work, we employed EC1, a reported RCD that can be activated by protein molecule secreted specifically by *E. coli* cells. To facilitate the attachment and functionality of EC1 on paper device, we used our previously proposed 3D DNA strategy for expedient engineering of functional DNA-based surfaces [30]. The device could detect *E. coli* K12 with a detection limit of 10^3^ CFU·mL^−1^. To investigate whether this paper sensor can specifically recognize *E. coli*, we selected two bacteria: (1) *Bacillus subtilis*, commonly found in soil, water sources and in association with plants, is the best-characterized member of the gram-positive bacteria [31]; (2) *Providencia sp.*, one common gram-negative bacteria, can cause diseases such as diarrhea, extraintestinal infections and urinary tract infections [32]. The good specificity of the device was demonstrated by the only positive result shown in the sample contaminated by *E. coli* whereas other samples (e.g., *Bacillus subtilis.* and *Providencia sp.*) showed negative results. This fully integrated oPAD holds great promise for future POC detection of pathogenic bacteria in resource-limited settings.

## 2. Materials and Methods

### 2.1. Oligonucleotides and Other Materials

All DNA oligonucleotides were obtained from Takara Biotechnology Co. (Dalian, China) and purified by standard 10% denaturing (8 M urea) polyacrylamide gel electrophoresis (dPAGE) or high-performance liquid chromatography (HPLC). T4 DNA ligase, dNTP, and T4 polynucleotide kinase (PNK) were purchased from Takara Biotechnology Co. (Dalian, China). ϕ29 DNA polymerase was acquired from Thermo Scientific (Waltham, MA, USA). Pullulan was obtained from Sangon Biotech (Shanghai, China). BCA Protein Assay Kit was acquired from Beyotime Biotechnology (Shanghai, China). Triton X-100 cell lysis buffer SS0890 was obtained from NOVON (Beijing, China). Nitrocellulose paper (HF180), Whatman filter paper 1# and Whatman 3MM CHR chromatography paper were purchased from GE Healthcare (Chicago, IL, USA). All other chemicals were purchased from Sangon Biotech (Shanghai, China) and used without further purification. Water was purified with a Greate Fun SUMMER-S-10 water purification system.

The sequences of the DNA molecules were as follows:

CDT1 (anti-EC1): GATATCATAT CACACAACTG TAAAGAAATC CATCCCCACA CAGTGTAGTG TTCCGGTCGC AGGTCTCGAC AACGCACATC (5′→3′)

TP1: ATATGATATC GATGTGCGTT (5′→3′)

DT1: ACGCACATCG ATATCATATC AC (5′→3′)

CDT2: TAGCTAGGAA GAGTCCCAAC CCGCCCTACC CAAAATGTCT CGGAT (5′→3′)

TP2: TTCCTAGCTA ATCCGAGACA (5′→3′)

F-RS28: ACTCTTCCTA GCFrAQGGTTC GATCAAGA-invert dT (5′→3′)

### 2.2. Instrumentation 

The fluorescent images of gels were obtained using a Typhoon 5 variable mode imager (GE Healthcare, Chicago, IL, USA) and analyzed using Image Quant software (Molecular Dynamics, GE Healthcare, Chicago, IL, USA). Paper well plates were printed using a Xerox ColorQube 8560N solid wax printer (Xerox Corporation, Norwalk, CT, USA), then heated at 120 °C for 2 min to melt the printed wax.

### 2.3. Synthesis of 3D EC1 via RCA

A circular DNA template (CDT1) containing anti-EC1 was first synthesized as previously described [30]. 1 μL of 10 μM CDT1 was mixed with 1 μL of 10 μM template primer (TP1), 5 μL of 10×RCA buffer, 5 μL of 10 mM dNTPs and 10 U of ϕ29DP, followed by incubation at 30 °C for 30 min. After heated at 90 °C for 5 min, the obtained RCA products were then incubated at room temperature for 12 h to self-assemble into 3D EC1. The resultant 3D EC1 were purified through a 3K membrane (NANOSEP OMEGA, Pall Incorporation) at 5,000 g for 10 min, and washed twice with pure water and collected.

### 2.4. Preparation of Bacterial Cells

A single colony of *E. coli* K12 grown on Luria Broth (LB) agar plate was taken and used to inoculate 50 mL of LB. After shaking at 37 °C until OD_600_ reached ~1, the bacterial culture was serially diluted in 10-fold intervals. 10 µL of each diluted solution were plated on the LB agar plates (3 repeats) and cultured at 37 °C for 18 h to determine the number of colony forming units per milliliter (CFU·mL^−1^) in the original bacteria cultures. Cells in the sample were then removed by centrifugation at 8000 g for 10 min at 4 °C. The crude intracellular mixture (CIM) from *E. coli* and *B. subtilis* cells (~10^7^ CFU·mL^−1^) was prepared according to our previously reported method [12,30].

### 2.5. Fabrication of oPAD

Each device contains four panels, namely: an absorbent pad (Panel A, 1.5 cm × 1.5 cm) for sample purification and washing; Whatman 3MM CHR chromatography paper (Panel B, with a diameter of 4 mm) for cell lysis; Whatman Grade 1# filter paper (Panel C, with a diameter of 4.5 mm) for DNAzyme recognition, and Nitrocellulose membrane HF180 (Panel D, with a diameter of 4.5 mm) for RCA. For dry reagent storage, 5 µL of cell Lysis Buffer was pipetted onto the Panel B. The Panel was dried in a vacuum oven at 50 °C for about 1 h. For Panel C, 5 μL of 3D EC1 was printed using a Biodot XYZ3060 automated dispensing unit (Somerville, MA, USA). After immersion into the blocking buffer (25 mM PBS containing 1% BSA, 0.02% Tween-20, pH 7.5) for 10 min, the obtained bioactive paper was dried at RT. 5 µL of F-RS28 (100 nM) was then mixed with 5 µL of 10% (*w*/*v*) pullulan, and then printed onto Panel C using the dispensing unit. For Panel D, 1 µL of 1 µM CDT2, 1 µL of 10 mM dNTPs, 1 µL of 100 µM hemin, 4 U of PNK and 5 U of ϕ29DP were first mixed with 20 µL of 10% (*w*/*v*) pullulan, and then printed. The bioactive paper was dried and stored at RT in a desiccant container. In order to assemble these four panels, a single sided adhesive film was used to enable bidirectional folding.

### 2.6. BCA (Bicinchoninic Acid) Protein Assay

5 μL of *E. coli* cells (10–10^7^ CFU·mL^−1^) was applied to the Panel B and incubated for 3 min. After washed by 50 μL of pure water, protein levels were measured with the BCA Protein Assay Kit according to the manufacturer’s protocol. The colorimetric result was recorded within ~1 min by a digital camera and analyzed by ImageJ.

### 2.7. Procedure for Bacterial Detection Assay

5 μL of *E. coli* cells (10–10^7^ CFU·mL^−1^) was applied to the Panel B. Following incubation for 3 min, 50 μL of pure water was then added. Panel B was then the folded onto Panel C and D to allow the flow of liquid. 50 μL of 1×RCA Buffer (33 mM Tris acetate, 10 mM magnesium acetate, 66 mM potassium acetate, 1% Tween-20, 1 mM, DTT, pH 7.9) was added to the back of Panel B to elute the protein. After the pullulan film was dissolved onto Panel D (~30 min), the flexible oPAD was folded to physically separate the Panels. 5 μL of 40 mM H_2_O_2_ and 5 μL of 20 mM TMB to initiate the colorimetric reaction on Panel D. The colorimetric result was recorded within ~1 min by a digital camera and analyzed by ImageJ.

## 3. Results and Discussions

### 3.1. Principle of Fully Integrated oPADs

The working principle and key functionalities of the oPADs are illustrated in Scheme 1. Panel B was first folded onto Panel A to enable the on-paper protein extraction from *E. coli*. By folding Panel B onto Panel C and Panel D, the purified protein targets were then transferred to Panel C, where the printed fluorogenic RNA substrate (named F-RS28, Takara Biotechnology Co., Dalian, China) was released, and the following 3D EC1-mediated RNA cleavage can be performed. The fluid containing 5’ cleaved fragment will further migrate to Panel D, where the printed pullulan-encapsulated RCA reagents (including ϕ29DP, CDT2, dNTPs, PNK and hemin) was dissolved and the following RCA reaction was activated using the 5’ cleaved fragments as new primers. The detection of *E. coli* is therefore easily converted to the detection of RCA product, which was designed to contain a repetitive unit of PW17, a peroxidase-mimicking DNAzyme capable of producing a colorimetric signal by oxidizing the chromogenic substrate TMB in the presence of hemin and H_2_O_2_ [30]. 

The overall operating processes include four steps: (1) cell loading and lysis. The impregnated dry lysis buffer is rehydrated on Panel B and provides the chemical lysis of *E. coli* cells; (2) protein purification. One wash to remove the potential inhibitors and surfactants on Panel B; (3) protein elution. One wash to elute the purified proteins from Panel B for the following DNAzyme-mediated cleavage on Panel C and RCA on Panel D; (4) signal readout. The chromogenic substrate and H_2_O_2_ were added to initiate the colorimetric reaction. The device can provide quantification of *E. coli* within 35 min, including 3 min of cell lysis, 30 min of cleavage reaction and RCA, and 1 min of signal read-out.

### 3.2. RNA Cleavage Activity of 3D EC1

To create the bioactive paper surface, we used the micron-sized 3D DNA superstructure to immobilize EC1 for the molecular recognition. 3D EC1 containing concatemeric EC1 was first synthesized by RCA. Transmission electron microscopy (TEM, FEI, Hillsboro, OR, USA) images in Figure 1a showed that monodispersed 3D EC1 was obtained with an average diameter of 2 μm. We then examined the activity of 3D EC1 using F-RS28 in which a single ribonucleotide is embedded in a DNA chain and is flanked by a pair of nucleotides labeled with FAM and Dabcyl. As shown in Figure 1b, 3D EC1 was indeed able to cleave F-RS28 in the presence of crude intracellular mixture (CIM) prepared from *E. coli*, resulting in a dramatic time-dependent increase in fluorescence. In contrast, no obvious fluorescence increase was observed upon the addition of *B. subtilis*. Taken together, these results indicated that 3D EC1 retained its high cleavage activity and specificity for *E. coli*.

### 3.3. On-Paper Protein Extraction

We then optimized the operation process of on-paper protein extraction. 5 µL of 10^6^
*E. coli* cells was pipetted onto Panel B containing dried lysis reagents. It took about 3 min for cell lysis (Appendix A). 50 μL of water was then flushed over Panel B two times in order to effectively remove the potential small inhibitors and surfactants. Several parameters may affect the performance of on-paper protein extraction. We first compared the protein extraction capabilities of several different types of papers, including GE nitrocellulose membrane (NC, 0.22 µm and 0.45 µm) and Whatman filter paper (Grade 1, 5 and 3MM). 10^6^
*E. coli* cells were pipetted onto the papers and washed with water, followed by BCA (Bicinchoninic acid) protein assay. As shown in Figure 2a, the Whatman Grade 5 filter paper provided the highest protein extraction efficiency. This result illustrated that the average pore size and the composition of the paper are related to the protein extraction efficiency. This on-paper extraction method could achieve the protein extraction from 10^3^ cells (Figure 2b). These results highlight its enticing advantage in terms of low sample volume, high yields and rapid processing.

### 3.4. Validation of the Assay

We then examined the DNAzyme-mediated RNA cleavage reaction on paper. Upon inkjet printing, 3D EC1 could retain its original spherical shape onto paper and adhere strongly to the paper surface via simple physisorption (Figure 3a). As shown in Figure 3b, a recovered fluorescence signal was observed when: (1) 3D EC1 was first printed on Panel C; (2) *E. coli* was provided and lysated on Panel B. 

We next tested the RCA reaction on paper. The cleaved RNA substrate will further migrate through the Panel C driven by capillary action. The 5’ cleavage fragment is expected to server as DNA primer for the RCA on Panel D, generating a PW17-containing RCA product, a peroxidase-like DNAzyme capable of generating a colorimetric signal. This expectation was confirmed experimentally (Figure 3; the others represent various controls). Note that T4 polynucleotide kinase (PNK) was needed to remove the terminal 2’,3’-cyclic phosphate in the 5’ cleavage fragment before the DNA polymerization.

### 3.5. Assay Performance

We then assessed the performance of the all-printed oPADs. Figure 4a shows the images of colorimetric responses of the oPADs in the presence of varying concentration of *E. coli*. The response curve was obtained by measuring the color intensity using ImageJ for each oPAD. One observed that the color intensity was proportional to the concentration of *E. coli* ranging from 10^3^ to 10^7^ CFU·mL^−1^. At 35 min, the oPAD can achieve a limit of detection (LOD) of 10^3^ CFU·mL^−1^. It was known that the colorimetric signal depended on the RCA product [29]. Increasing the RCA reaction time may lead to increasing amounts of RCA products, thus further improving the sensitivity. The selectivity was also tested against two gram-negative bacteria (*B. subtilis.* and *Providencia sp.*). These control bacteria were not able to induce a significant color change, indicating the high specificity of our device (Figure 4b).

To verify the potential application of our fully disposable and integrated oPAD, we spiked *E. coli* (10^7^ CFU·mL^−1^) into different samples, including milk and juice. It was observed that this device was able to produce a positive signal (Figure 4c). These results suggest that our assay could be used for *E. coli* detection in complex media.

## 4. Conclusions

In summary, we developed a fully disposable and integrated paper-based device by integrating paper-based protein extraction, paper-based FNAs sensor, paper-based isothermal DNA amplification and portable readout. The detection limit of our device was 10^3^ CFU·mL^−1^. Our assay required no instrumentation other than a smart phone, and the total sample-to-answer assay time was 35 min, making it suitable for POC testing. As a comparison, the detection limit of one commonly used pathogen detection method, sandwich enzyme-linked immunosorbent assay (ELISA), is approximately 10^3^–10^4^ CFU·mL^−1^ in several hours [33]. Thus, our sensor offers a comparable detection sensitivity, while the test time was significantly shortened. With a wide variety of DNAzymes currently available and new DNAzymes that can be identified by in vitro selection, we envision that the described device here would be a powerful tool for pathogen testing in resource-limited settings. To further improve the universality of this paper-based device, our ongoing work would focus on extracting proteins from various patient samples such as blood and stool, which can further expand the application of this device at the POC diagnostics. Moreover, methods for actually detecting pathogenic bacteria (such as *E. coli*) require detection limit of as few as 1–100 CFU in food or water. Thus the challenge that must be addressed is how to make this paper device suitable for integration with sample enrichment for real-world application.

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
