# Peer review of "An Origami Paper-Based Device Printed with DNAzyme-Containing DNA Superstructures for *Escherichia coli* Detection"

_micromachines, 2019, doi:10.3390/mi10080531_

Round 1
Reviewer 1 Report
In this manuscript, the authors reported on an Origami paper-based device for the detection of E. coli. The research is innovative in a way that it combines cell lysis, RNA-cleaving DNA, RCA, and Origami technique for the development of the device. The manuscript is well written, the results are sufficient and analysed adequately. The present results indicate that the device was working for the detection of E. coli at the concentration of as low as 10E3 CFU mL-1. The conclusion is well drawn based on the observed data. Given the innovative aspect and the sound result, I would like to suggest a publication as is.
Author Response
We greatly appreciate the referee’s positive comments.
Reviewer 2 Report
The manuscript by Yating Sun et al. describes the development a of a low cost and rapid paper-based biosensor for point-of-care testing to detect specific bacterial pathogens, including proof-of-concept experiments to specifically detect E. coli compared to other bacteria and in complex media. New and of great interest is the integration of different biochemical steps in one analytical device, combined with the new use of DNAzymes that simplifies the production and usage. However, the manuscript suffers from a lack of some information and some more detailed explanations here and there. I recommend publication after a minor revision addressing the following issues and questions:
Introduction:
In the first paragraph, the given information about diseases based on bacterial pathogens is a bit confusing. For example, why were examples given for Canada only? The following children death information is worldwide, but seems at first from the text that it refers to Canada. Since the introduction starts with bacterial pathogens, and the development is aiming to detect those, why were mixed information given for viruses? I would recommend here to i) focus on bacterial pathogens only (maybe adding some more of typical ones that cause diseases worldwide, e.g. Cholera, Campylobacter, Listeria,…), and ii) give statistics either for several countries, or only for worldwide incidents. There is also no information about the chosen control bacteria used later in the experiments. Why were they chosen and how do they impact human health, if they do?
Results & Discussion:
1. Scheme 1: The schematics would benefit with more legends and some brief explanations in the caption.
2. I was wondering how the 3D DNA adheres to the filter paper. While the creation of 3D DNA was described in a previous publication, it would be nice to add a few sentences about the adhesion strategy and strength of 3D DNA on the filter paper.
3. In line 186, the authors state that cell lysis takes ~ 3 min. There is no reference or data whatsoever. Could you add one or the another?
4. BCA protein assay was used to determine protein amount in dependency of different filters used. The method is not described in the material and methods section. Please add, even though it was a commercial kit.
5. In chapter 3.3 the reference to Figure 2b is missing. Please add.
6. Line 193-195: in the text it was stated that 103 CFU/ml already provided sufficient proteins, there is not statistical test (e.g. t-test or ANOVA) that supports this statement. From only looking at the data, I could argue that it lies within the error bars with lower cell concentrations. Same holds for the data shown in Figure 2a; the differences seem non-significant. Please provide statistical tests.
7. In line 220, the authors state that the oPAD can achieve its highest sensitivity after 35 min. The figure 4, however, only shows the detection efficiency in dependence on the concentration. Would a longer time increase the probability to detect reproducibly also lower concentrations (e.g 102 CFU/ml)? These questions can be avoided and answered by adding the mentioned but not shown response curves.
8. Since B. subtilis was used to verify the selectivity of 3D EC1 (Figure 1b) to E. coli, why was B. subtilis not used as control together with the other control bacteria? Since the chosen control bacteria are not fully attributed to diseases, why were these chosen? The use of B. subtilis would be an example of an actual mild pathogen, comparable to E. coli.
Conclusion:
1. The authors state a LOD of 103CFU/ml. How is this performance compared to those of other, similar biosensors? Is this sensitivity sufficient enough for real-world application?
2. What are the remaining challenges for the application in real world? E.g. test expiry time, which improvements of sensitivity, reliability, etc.
Experimental data and statistics:
- Figure 1b: I assume that only one experiment was performed. It would be necessary to see the reproducibility of this procedure by averaging ≥3 repetitions.
- Figure 2: Please state how many experimental repetitions were made, what the error bars represent (e.g. SD or SEM), and statistical tests (e.g. t-test or ANOVA) in respect to the stated result in the according text.
- Figure 3: Please add colorimetric values to the Panel D images, since the differences are not clearly visible.
- Figure 4a: Please state how many experimental repetitions were made, what the error bars represent (e.g. SD or SEM). Add in the figure legend and caption the shown fit type and fit quality value.
- Figure 4b,c: Please add colorimetric values to the Panel D images, since the differences are not clearly visible in all cases.
Language:
Please revise the text in respect to some misspellings and incomplete phrases.
Round 2
Reviewer 2 Report
Dear authors,
Your revised manuscript version notably improved in content and quality by providing all previously missing information, clarification, and additional data. The initial objective of this manuscript is now reached and I can recommend its publication in the present form.
Keep up the good work.